# Impact of tiered restrictions on human activities and the epidemiology of the second wave of COVID-19 in Italy

Mattia Manica [1,2], Giorgio Guzzetta[1,2], Flavia Riccardo [3], Antonio Valenti[4], Piero Poletti [1,2], Valentina Marziano [1,2], Filippo Trentini [1,2], Xanthi Andrianou[3,5], Alberto Mateo-Urdiales [3,6], Martina del Manso[3,6], Massimo Fabiani[3], Maria Fenicia Vescio [3], Matteo Spuri[3], Daniele Petrone[3], Antonino Bella[3], Sergio Iavicoli[4,9], Marco Ajelli [7,8,9], Silvio Brusaferro[3,9], Patrizio Pezzotti [3,9] & Stefano Merler [1,2,9 ✉]

To counter the second COVID-19 wave in autumn 2020, the Italian government introduced a system of physical distancing measures organized in progressively restrictive tiers (coded as yellow, orange, and red) imposed on a regional basis according to real-time epidemiological risk assessments. We leverage the data from the Italian COVID-19 integrated surveillance system and publicly available mobility data to evaluate the impact of the three-tiered regional restriction system on human activities, SARS-CoV-2 transmissibility and hospitalization burden in Italy. The individuals' attendance to locations outside the residential settings was progressively reduced with tiers, but less than during the national lockdown against the first COVID-19 wave in the spring. The reproduction number R(t) decreased below the epidemic threshold in 85 out of 107 provinces after the introduction of the tier system, reaching average values of about 0.95-1.02 in the yellow tier, 0.80-0.93 in the orange tier and 0.74-0.83 in the red tier. We estimate that the reduced transmissibility resulted in averting about 36% of the hospitalizations between November 6 and November 25, 2020. These results are instrumental to inform public health efforts aimed at preventing future resurgence of cases.

[1] Center for Health Emergencies, Bruno Kessler Foundation, Trento, Italy. [2] Epilab-JRU, FEM-FBK Joint Research Unit, Trento, Italy. [3] Istituto Superiore di Sanità, Rome, Italy. [4] Department of Occupational and Environmental Medicine, Epidemiology and Hygiene, Italian Workers' Compensation Authority (INAIL), Rome, Italy. [5] Cyprus University of Technology, Limassol, Cyprus. [6] European Programme for Intervention Epidemiology Training (EPIET), European Centre for Disease Prevention and Control (ECDC), Stockholm, Sweden. [7] Department of Epidemiology and Biostatistics, Indiana University School of Public Health, Bloomington, IN, USA. [8] Laboratory for the Modeling of Biological and Socio-technical Systems, Northeastern University, Boston, USA. [9] These authors jointly supervised this work: Sergio Iavicoli, Marco Ajelli, Silvio Brusaferro, Patrizio Pezzotti, Stefano Merler. ✉email: merler@fbk.eu

The second wave of COVID-19 has been spreading in all European countries in the fall of 2020[1]. In Italy, the daily incidence of confirmed cases rose slowly from 2 to 3 per 100,000 over the month of September, and then accelerated rapidly in October reaching a peak of 58 per 100,000 by November 13[1]. The second wave resulted in about 1.2 COVID-19-related deaths per 100,000 per day at the beginning of December; a value comparable to the first wave (1.35 per 100,000). The mortality rate has then declined to a stable plateau of about 0.8 deaths per 100,000 per day throughout the month of January 2021[1,2]. At the subnational level, high geographical heterogeneity in the impact of the second wave was observed, with over four-fold variations across regions in the standardized mortality rate in October and November 2020[3].

To counter the rapid rise in SARS-CoV-2 infections observed since the end of September, the Italian government has progressively increased restrictions aimed at promoting physical distancing[4–7]. Between October 14 and November 5, 2020, interventions were uniformly enacted at the national level. These measures initially extended the mandatory use of face masks to outdoor spaces (previously mandated only indoors) and targeted a reduction of opening hours of bars and restaurants as well as a reduction of the capacity of recreational venues such as cinemas and theaters. Shortly after, recreational venues and sports centers were closed altogether, and distance learning for at least 75% of the time was introduced in high schools. Starting from November 6, a three-tiered restriction system was introduced on a regional basis. Tiers were assigned to each of the 21 regions and autonomous provinces (AP) after an epidemiological risk assessment by the Ministry of Health, based on the combination of 21 quantitative indicators on (i) the level of transmission, (ii) the burden on older age groups and healthcare, and (iii) resilience of monitoring systems[7]. The sets of measures in the three tiers were labeled according to a color scheme: yellow, orange, and red, corresponding to increasing levels of restrictions. The tiered measures involved further limitations to retail and service activities, individual movement restrictions (ranging from a curfew between 10 p.m. and 5 a.m. to a full-day stay-home mandate with a ban on inter-regional mobility), and reinforced distance learning in schools (see Table 1 for a complete list). Due to the intrinsic delays in the dynamics of infection, healthcare seeking, and data collection, estimates of the reproduction number (measuring the level of transmission) available at each epidemiological assessment were relative to 14 days before the date of assessment. To guarantee a prompt reaction to epidemiological updates, the assignment of tiers was therefore re-evaluated according to frequent reassessments of incoming data from regions.

The assessment of the effectiveness of the adopted interventions is critical to guide future decisions for the management of COVID-19 in Italy as well as other countries. In this work, we leverage the data from the Italian COVID-19-integrated surveillance system and publicly available mobility data[8] to evaluate the association of the three-tiered regional restriction system with changes in human activities, SARS-CoV-2 transmissibility, and hospitalization burden in Italy.

## Results

### Changes in human activities after the introduction of tiers.
Before the progressive introduction of national restrictions on October 14, 2020, mobility indicators were stable in all categories (Fig. 1). The progressive implementation of restrictions with spatially and temporally heterogeneous applications (see Table 1) between October 14 and November 6 (day of introduction of tiers) induced a clear trend of decline of human activities, with a decrease in the number of visitors in locations outside the residential settings and an increase in the time spent at home (Fig. 1). The introduction of tiers was followed by a further brisk reduction in human activities, which was higher where more restrictive tiers were adopted. After November 6, indicators remained substantially stable, except for a possible reduction of compliance in the red tier (Fig. 1). The mobility indicators most impacted by tiers were those related to retail and recreation activities, as well as public transportation means (Table 2), where an over 50% reduction of presence on average was associated with the red tier with respect to pre-pandemic values (January 5–February 6, 2020). The decline of attendance rates in these settings was mirrored by an increase of the time spent at home from 6.9% (95% CI: 4.2–10.6%) above the pre-pandemic values before the setup of tiers, to up to 17.7% (95% CI: 15.8–20.6%).

### Changes in transmissibility.
We considered changes in transmissibility at the regional and provincial levels. The regional level comprises 19 regions plus the Autonomous Provinces (APs) of Trento and Bolzano, while the provincial level comprises 107 provinces (including the two APs). As a measure of transmissibility, we considered the net reproduction number $R(t)$, estimated from the epidemic curve of symptomatic cases by date of symptom onset, given a distribution of the serial interval. The temporal dynamics of the net reproduction number $R(t)$ in November 2020 were highly variable both at the regional and provincial levels (Fig. 2). The observed net reproduction number between November 19 and 25 fell below the epidemic threshold in 42 of 46 (91%) provinces having the red tier as the maximum restriction, in 33 of 41 (81%) provinces having the orange tier as the maximum restriction and only in 10 of 20 provinces that remained in the yellow tier over the study period (50%), despite the latter starting from much lower $R(t)$ values (see Supplementary Fig. 3).

We applied a number of models (Table 3; details in the section "Methods") to quantify the association between tiers and the change of $R(t)$ between the week October 30–November 5 (before the application of tiers) and the week November 19–25 (after the effect of tiers on $R(t)$ have stabilized). The yellow tier was associated with a mean net reduction of $R(t)$ ranging between 0.15 and 0.23 (Fig. 3A, B; a range of CIs: 0.06–0.34), corresponding to a relative reduction between 13% and 19%, in all linear mixed models (LMMs) except when using regional rather than provincial data (Model D): in this case, the mean reduction was estimated to be lower (net 0.09, relative 7%) and with broad confidence intervals due to the limited number of data points. The orange tier was associated with a mean reduction in $R(t)$ ranging between 0.34 and 0.52 (range of 95% CIs: 0.27–0.65) across all LMMs, corresponding to a relative reduction between 27% and 38%. The red tier was associated with a mean reduction in $R(t)$ ranging between 0.47 and 0.63 (range of 95% CIs: 0.41–0.71) across all LMMs, corresponding to a relative reduction between 36% and 45%. In the week November 19–25, the modeled mean $R(t)$ was in the range 0.97–1.02 in the yellow tier, 0.81–0.93 in the orange tier and 0.74–0.83 in the red tier (Fig. 3C). All models considered for Fig. 3 are based on a grouping of provinces or regions by tiers. To avoid potential biases associated with the grouping, we applied an additional model (Model F in Table 3) that estimates the expected final $R(t)$ resulting from the application of a given tier while taking into account regional changes over time in the assigned tier (see the "Methods" section). This model estimated a final value for $R(t)$ of 0.95 (0.94–0.97) in the yellow tier, 0.80 (0.78–0.82) in the orange tier, and 0.77 (0.76–0.78) in the red tier, in good agreement with estimates from LMMs.

**Table 1 Description of restrictions applied in Italy since October 14[4–7,16–20].**

| Restrictions | National restrictions (October 14–November 5, 2020) | November 6, 2020-onwards | | |
|---|---|---|---|---|
| | | Yellow tier | Orange tier | Red tier |
| Face masks | Mandatory in outdoor spaces | Mandatory in outdoor spaces | Mandatory in outdoor spaces | Mandatory in outdoor spaces |
| Individual movements | No restrictions | Stay-home mandate between 10 p.m. and 5 a.m. (except for work, health and other certified reasons) | Stay-home mandate and ban on movements between municipalities and to/from other regions (except for work, health and other certified reasons) | Stay-home mandate and ban on movements between municipalities and to/from other regions (except for work, health and other certified reasons). |
| Retail and services | Open | Shopping malls closed during weekends and holidays (with the exception of essential retail & services) | Shopping malls closed during weekends and holidays (with the exception of essential retail & services) | All shops closed (with the exception of essential retail & services) |
| Schools & childcare | Open until October 18. Recommendation to adopt distance learning for high schools and universities since October 19. Mandatory distance learning for at least 75% of the time in high schools since October 26. Regional exceptions:- in Campania, kindergartens were closed and distance learning for all schools adopted since October 16; - in Apulia, distance learning for all schools adopted since October 30; - in Lombardy, 100% distance learning for high schools since October 26; - in Calabria, 100% distance learning for high schools and universities since October 26. | Distance learning in high schools and universities except when on-site attendance is essential (i.e., for laboratory activities) | Distance learning in high schools and universities except when on-site attendance is essential (i.e., for laboratory activities) | Distance learning in second and third grade of middle schools, in all grades of high schools and universities |
| Bars serving food, Cafés & Restaurants | No service after 12 a.m. until October 25. No service after 6 p.m. and take away allowed until 12 a.m. since October 26. | No service after 6 p.m. and take away allowed until 10 p.m. | Closed. Take away allowed until 10 p.m. | Closed. Take away allowed until 10 p.m. |
| Public transport | No capacity reduction. In Umbria, 50% capacity reduction since October 21. | 50% capacity reduction (except school service) | 50% capacity reduction (except school service) | 50% capacity reduction (except school service) |
| Indoor recreational and cultural venues | Open with capacity reduction until October 25. Closed since October 26. | Closed | Closed | Closed |
| Gyms, pools & leisure venues | Open until October 25. Non-professional contact sports not permitted. Closed except outdoor sport centers since October 26. | Closed except outdoor sport centers | Closed except outdoor sport centers | Individual outdoor training only (except sports events of national interest) |

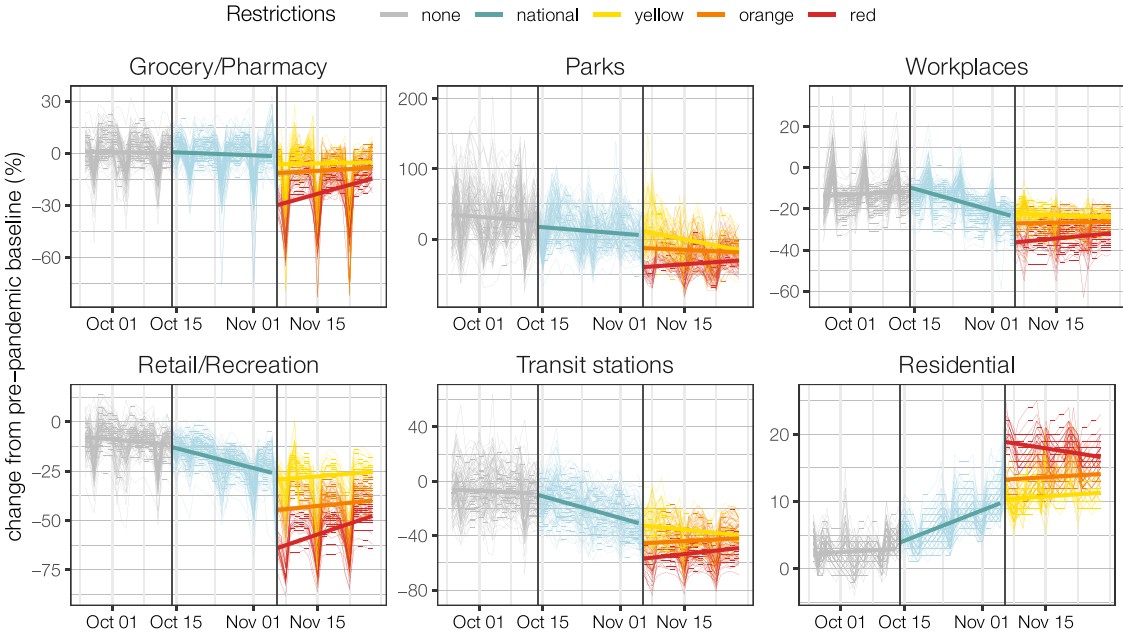

**Fig. 1 Changes in the time spent in different locations relative to pre-pandemic values over time[8], September 25–November 25.** Colors represent three distinct intervention periods, separated by vertical darker lines: September 25–October 13 (gray, no interventions), October 14–November 5 (teal, national interventions), November 6–November 25 (yellow, orange and red, aggregating provinces by the corresponding tier). Thin lines represent values for individual provinces. Thick lines are regression lines computed over provinces with the same restrictions and are reported to help the visual identification of trends in different periods.

**Table 2 Change in the number of visitors in different locations relative to pre-pandemic values, as estimated by a linear mixed model (mean and 95%CI, values in percentage).**

| Location | National | Yellow | Orange | Red |
|---|---|---|---|---|
| Grocery/Pharmacy | −0.9 (−2.7; 0.9) | −6.2 (−8.0; −4.3) | −12.2 (−14.1; −10.3) | −22.0 (−24.0; -20.1) |
| Parks | 11.1 (7.3; 14.9) | 5.0 (0.8; 9.1) | −20.1 (−24.3; −15.8) | -34.1 (−38.6; -29.6) |
| Retail/Recreation | −20.8 (−23.5; −18.2) | −29.8 (−32.5; −27.1) | −46.6 (−49.3; −43.8) | −55.1 (−57.9; -52.3) |
| Transit stations | −19.7 (-22.6; −16.7) | −34.4 (−37.4; −31.3) | −44.6 (−47.7; -41.5) | −50.9 (−54.1; −47.7) |
| Workplaces | −16.7 (−17.9; −15.4) | −23.4 (−24.7; −22.1) | −28.2 (−29.6; −26.9) | −32.6 (−34.0; −31.2) |
| Residential | 6.9 (6.4; 7.3) | 10.6 (10.1; 11.1) | 14.7 (14.2; 15.2) | 16.9 (16.4; 17.4) |

For residential locations, the reported value refers to the changes in the time spent. National interventions were in place from October 14 to November 5, 2020, while values for different tiers refer to the period November 6-25.

**Changes in hospital admissions**. We compared the observed values of the daily hospital admissions with projections obtained under the assumption that the tier system was not introduced on November 6, i.e., that $R(t)$ remained constant at the value of October 30–November 5, 2020 ("status quo" scenario). The observed cumulative incidence of hospital admissions over the study period was 62.7, 66.3, and 83.8 per 100,000 in regions with maximum tier yellow, orange and red, respectively, corresponding to 44,350 hospitalizations in the whole country (Fig. 4). If $R(t)$ estimated at November 5 had remained unaltered until November 25, we estimate a cumulative incidence of hospital admissions of 74.7 (95% CI: 71.1–78.8), 105.3 (95% CI: 101.4–109.4), and 139.5 (95% CI: 135.6–143.1) per 100,000 in regions with maximum tier yellow, orange and red respectively, resulting in a total of 68,880 (95%CI: 67,590–70,335) hospital admissions. Thus, we estimate that about 35.6% (95% CI: 34.4–36.9%) of the overall hospitalizations in the considered period (i.e., about 24,500) were avoided, with a reduction of 15.9% (95% CI: 11.8–20.4%) in the yellow tier, 37.0% (95% CI: 34.6–39.3%) in the orange tier and 41.4% (95% CI: 39.9–41.5%) in the red tier.

**Discussion**

The three-tiered restriction system introduced on a regional basis by the Italian government on November 6, 2020, was associated with significant changes in both human activities and SARS-CoV-2 transmission. As of May 2021, the tier system is still in place, with limited changes to the enacted restrictions, and represents the key strategy used by the Italian government to control and mitigate SARS-CoV-2 transmission as vaccination rolls out. For what concerns human activities, we found a significant and progressive reduction of the time spent outside of the home in all locations recorded by the Google mobility data[8], especially those associated with recreational and retail activities, and public transport. This is not surprising, considering that the restrictions mainly acted on social gathering venues that had not been targeted by previous interventions, such as bars, restaurants and shopping malls, and that they introduced limitations to individual movements. Reductions in attendance in schools (data not available in the Google mobility data) and workplaces may have contributed to the reduction in public transport use. We found that the activity reduction in all locations outside of the home was

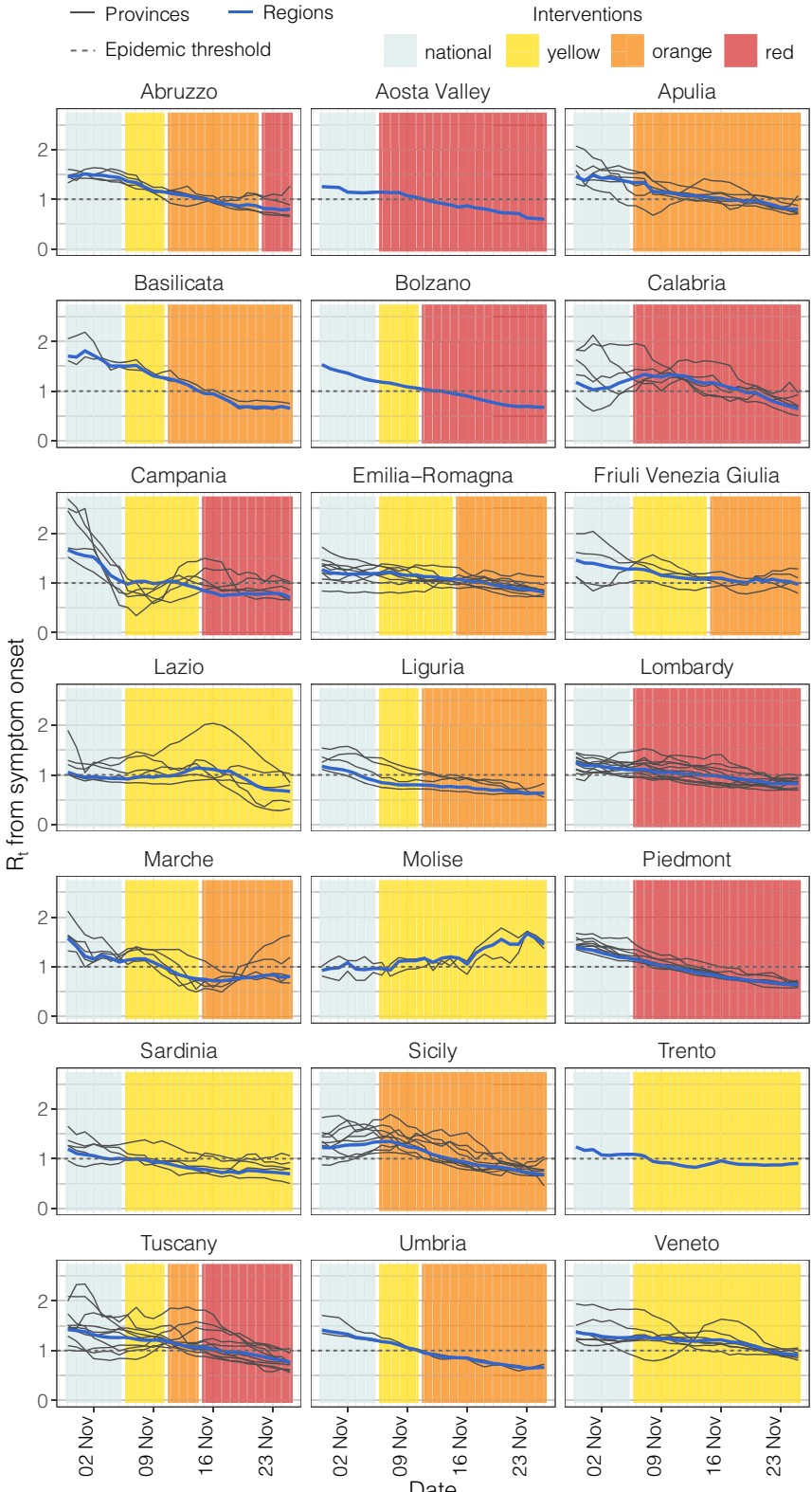

**Fig. 2 Temporal dynamics of the net reproduction numbers *R(t)* and of the assigned tiers between October 30 and November 25.** Each line shows the mean *R(t)* for an Italian province (black) or region (blue). Provinces are grouped by region as tiers were assigned on a regional basis. Colored rectangles refer to the timeframe when the different tiers were adopted: teal = national interventions, yellow, orange and red as the corresponding tier (see Table 1 for restrictions associated with the different tiers).

**Table 3 Models to evaluate the change in transmissibility ($R(t)$) associated to tiers.**

| Model | Method | Mean si (days) | Geographic scale | Number of data points | Estimated outcome | Grouping | $R^2$ |
|---|---|---|---|---|---|---|---|
| A | LMM | 6.68 | Province | 214 | Net change | Maximum tier | 0.58 |
| B | LMM | 6.68 | Province | 214 | Relative change | Maximum tier | 0.62 |
| C | LMM | 6.68 | Province | 214 | Net change | Tier | 0.58 |
| D | LMM | 6.68 | Region | 42 | Net change | Maximum tier | 0.70 |
| E | LMM | 5.01 | Province | 122 | Net change | Maximum tier | 0.56 |
| F | Renewal equation | 6.68 | Region | 21 | Final $R(t)$ | No grouping | – |

Net change refers to the difference between the mean $R(t)$ in the period October 30–November 5, 2020 and the mean $R(t)$ in the period November 19–25, 2020. Relative change refers to the net change divided by the mean $R(t)$ in the period October 30–November 5, 2020. Final $R(t)$ refers to the expected final value of $R(t)$ after 2 weeks since the enactment of the tier, considered as the time for an intervention to reach its full effect on $R(t)$[11]. Maximum tier indicates that provinces and regions were grouped by the strictest tier enacted over the study period. In model C, only provinces that did not change tier over the study period were selected, and the assigned tier was used for the grouping. $R^2$ is the estimated marginal coefficient of determination computed based on Nakagawa et al.[27].
LMM linear mixed model, SI serial interval.

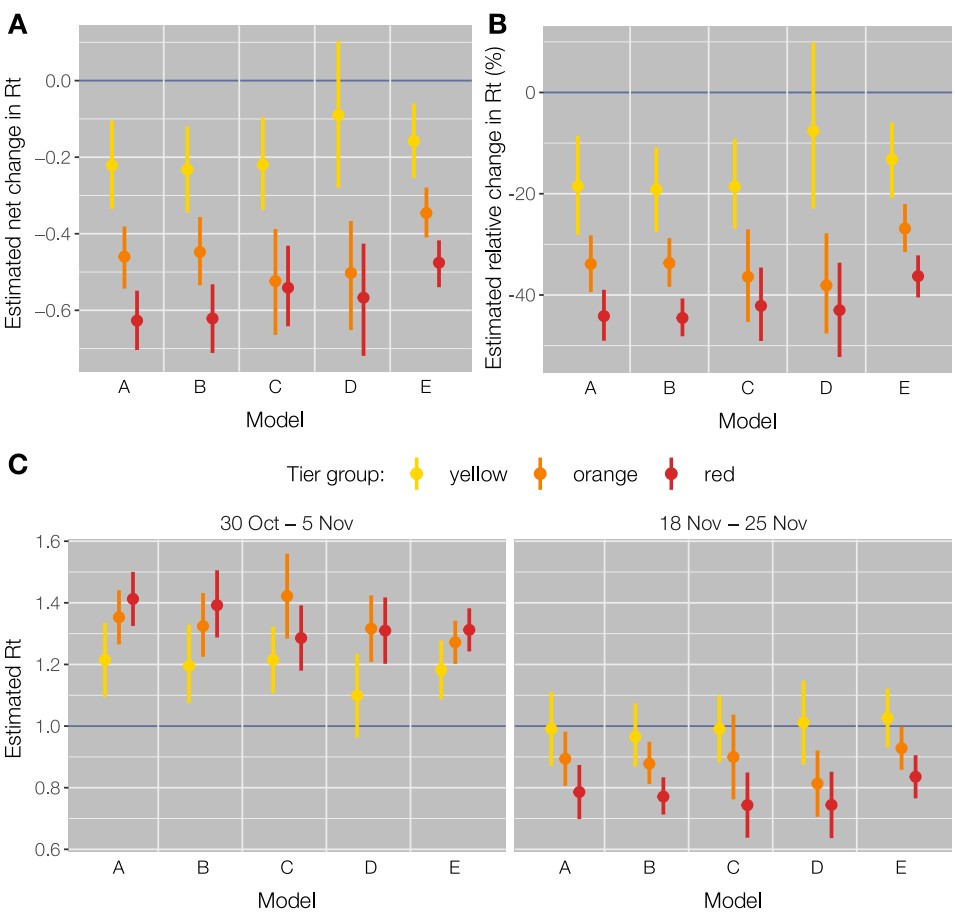

**Fig. 3 Estimates from LMMs. A** Net change of the reproduction number between the week October 30–November 5, 2020, and the week November 19–25, 2020, by tier group. $n = 107$ observations (provinces) observed over 2 time periods for model A, B,C, and E, $n = 21$ observations (regions) observed over 2 time periods for model D. **B** Relative change of the reproduction number between the week October 30–November 5, 2020, and the week November 19–25, 2020, by tier group, $n = 107$ observations (provinces) observed over 2 time periods for model A, B, C, and E, $n = 21$ observations (regions) observed over two time periods for model D. **C** Mean reproduction number in the week October 30–November 5, 2020, and in the week November 19–25, 2020, by tier group, $n = 107$ observations (provinces) in each time period for model A,B,C, and E, $n = 21$ observations (regions) in each time period for model D. Dots (center of the error bars) represent the mean values, vertical lines represent 95% CI. Colors (yellow, orange and red) are associated with the corresponding tier group. See Table 3 for a description of models.

far from that observed during the nationwide lockdown imposed to counter the first wave, even in the strictest tier where a stay-home mandate was in place. As a comparison, during the lockdown, the time spent in retail/recreational locations and in transit stations had dropped by over 75% with respect to pre-pandemic values (against 50% in the red tier, Table 2), and the time spent at home had increased by 27% (against 17% in the red tier)[8].

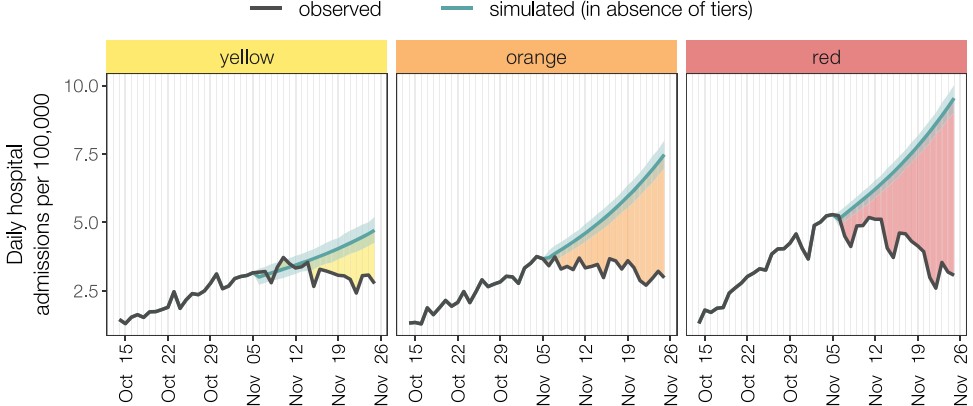

**Fig. 4 Incidence of hospital admissions by tier level over the period October 14–November 25, 2020.** Regions are grouped by the maximum tier assigned over the study period. Black lines represent observed hospital admissions. Teal lines represent mean projected values (shaded area: 95% projection interval) under the assumption that national restrictions were maintained after November 6, i.e., that the reproduction number was constant over the projection period. Shaded areas between the two curves highlight averted hospitalizations, the colors of each shaded area are associated with the corresponding tier.

On the epidemiological side, we found that reproduction numbers were close to 1 at the end of the study period in the yellow tier, while in orange and in red tiers the reproduction number was significantly below the epidemic threshold. Overall, provinces in the yellow tier achieved a mean reduction of $R(t)$ of 0.15–0.23 (13–19%) with respect to the transmissibility determined by the preceding nationwide restrictions, while a reduction of 0.34–0.52 (27–38%) was estimated in the orange tier and of 0.47–0.63 (36–45%) in the red tier. These results remained consistent across a number of models, which considered alternative choices on the target variable to be estimated (net vs. relative change of $R(t)$ vs. final $R(t)$), on the level of geographic aggregation (province vs. region), on the selection of regions (all regions, grouped by maximum tier vs. only regions that did not change tier over the study period), and on the duration of the serial interval (Table 3). At the national level, we estimate that the reduction in transmissibility averted about 24,500 hospital admissions between November 6 and 25, 2020, with larger gains in regions assigned to stricter tiers. We note, however, that the benefits of the introduction of tiers on the hospital burden extend well after November 25, thanks to reductions observed in the daily hospitalization incidence in all tiers (Fig. 4).

It is worth remarking that during periods of high infection incidence, the notification rate may decrease due to the saturation of tracing and testing capabilities. Although the method we adopted leads to estimates of $R(t)$ that are not affected by a constant underreporting rate[9], we acknowledge that fluctuations in the reporting rate may alter $R(t)$ estimates[10–12]. However, during the second wave of COVID-19 in Italy, the largest increase in the number of cases occurred in October; therefore, we expect the notification rate to have stabilized before the period considered for $R(t)$ (October 30–November 25). In addition, hospital admission rates are less subject to changes compared with notification rates of symptomatic cases, and we found similar results when using transmissibility estimates computed from the curve of hospital admissions (see Supplementary Tables 14 and 15). It is important to consider that a part of the estimated reduction in reproduction numbers has resulted from the accumulation of natural immunity. Between November 6 and 25, 2020, the cumulative notified cases represented 1.1% of the total Italian population. Considering a case notification ratio of 37%[13], we estimate a 3% depletion of the susceptible reservoir in Italy (with a range of 1–5% across Italian regions), translating into similar reductions of $R(t)$ due to the accumulated immunity. We also

note that nationwide restrictions implemented to counter the second wave were scaled up on three different occasions (on October 14, 19, and 25[4–6]) before adopting the three-tiered regional system since November 6, 2020[7]. It is, therefore, possible that part of the decrease of $R(t)$ after November 6 is associated to a residual effect of earlier interventions. However, previous studies have shown that most of the reduction in $R(t)$ takes place within about 2 weeks after the introduction of restrictions[12]. Therefore, this limitation should not have a major effect on our conclusions. The proposed analysis is not suitable to pinpoint which specific restrictions maximized the reduction in transmissibility[14,15], to disentangle the effect of spontaneous behavioral changes, and to capture possible cross-regional effects. For example, provinces in the yellow tier sharing borders with regions in the orange or red tier may have indirectly benefited from a reduction of inter-regional mobility or from a higher inclination of residents to self-impose restrictions to their activity patterns. Finally, our estimates are subject to further potential biases related to the ecological study design; therefore, figures associated with each tier should be taken with caution, despite their robustness across a number of modeling assumptions. For example, regions that experienced an escalation in their assigned tier over the study period might have altered our estimates for the effect of lower tier(s) if they were not reassigned, thus potentially biasing upward the estimated reduction in $R(t)$.

We quantified the epidemiological effect of the three-tiered system of restrictions adopted in Italy on a regional basis. We showed that stricter restrictions (orange and red tiers) were associated with a decreasing incidence (below-threshold reproduction numbers) and that the most permissive tier (yellow) was sufficient to reduce the reproduction number to values close to the epidemic threshold. We also showed that the tier system resulted in a much lower impact on human activities compared to lockdown and in large reductions in daily hospitalizations. These insights can help support efforts to control the incidence of COVID-19.

## Methods

**Restrictions data.** We collected information from official sources on the measures taken by the Italian government between October 14[4–7,16–20] and November 25. Eleven of 21 regions and APs maintained the same tier from November 6 throughout the study period; for all remaining regions except Abruzzo, the highest tier corresponded also to the one which has been maintained for the longest time (see Fig. 1).

**Mobility data**. We retrieved data from the Google community mobility reports at the provincial level[8] over the period September 25–November 25, 2020. These data represent the daily number of visitors at different locations, normalized to the value computed between January 5 and February 6, 2020 (pre-pandemic value). The locations reported in the data are categorized as follows: Grocery/Pharmacy (grocery markets, food warehouses, farmers markets, specialty food shops, drug stores, and pharmacies); Parks (local parks, national parks, public beaches, marinas, dog parks, plazas, and public gardens); Transit stations (public transport hubs such as subway, bus, and train stations); Retail/Recreation (restaurants, cafes, shopping centers, theme parks, museums, libraries, and movie theaters); and Workplaces (places of work). In addition, human activity in Residential (places of residence) is reported in terms of the mean duration of stay in these locations.

**Epidemiological data**. Data to estimate the reproduction numbers and hospital admissions were collected by regional health authorities and collated by the Istituto Superiore di Sanità (Italian National Institute of Health) within an integrated surveillance system (described in ref. [21]). As a measure of transmissibility, we considered the net reproduction number $R(t)$[10–12,22]. The posterior distribution of $R(t)$ at any time point $t$ was estimated by applying the Metropolis–Hastings MCMC sampling to a likelihood function defined as follows:

$$\mathscr{L} = \prod_{t=1}^{T} P\left(C(t); R^*(t) \sum_{s=1}^{t} \varphi(s) C(t-s)\right) \qquad (1)$$

where

- $P(k; \lambda)$ is the probability mass function of a Poisson distribution (i.e., the probability of observing $k$ events if these events occur with rate $\lambda$).
- $C(t)$ is the daily number of new cases having symptom onset at time $t$;
- $R^*(t)$ is the net reproduction number at time $t$ to be estimated;
- $\varphi(s)$ is the integral of the probability density function of the generation time evaluated between day $s-1$ and s.

We computed $R^*(t)$ for each of the 107 Italian provinces and for the 21 regions/APs, using, as a proxy for the distribution of the generation time, the distribution of the serial interval estimated from the analysis of contact tracing data in Lombardy[23] (a gamma function with shape 1.87 and rate 0.28, for a mean of 6.68 days). The values of $R(t)$ used throughout this study are then computed as the weekly moving average of $R^*(t)$.

**Changes in human activities after the introduction of tiers**. To assess the impact of tiers on human activities, we used mobility data to calibrate a linear mixed model for each of the location categories in the Google reports[8]. The LMMs are of the form:

$$M_{p,l} = \beta_0 + \beta_1 L_p^{\text{yellow}} + \beta_2 L_p^{\text{orange}} + \beta_3 L_p^{\text{red}} + a_r + b_{r,p} + \varepsilon_{p,T} \qquad (2)$$

where

- $M_{p,l}$ represents the mobility value of a given location category (i.e., the change in the number of visitors in non-residential locations, or the change in the time spent at home, normalized to the pre-pandemic values) in each of the 107 Italian provinces ($p$), averaged over the days in which a given tier $l$ was enforced;
- $L_p^l$ is a binary variable set to 1 when the considered value $M_{p,l}$ belongs to a province with tier $l$, and 0 otherwise;
- $\beta_0, \beta_1, \beta_2$ and $\beta_3$ and are model parameters, with $\beta_0$ representing the mean mobility across Italian provinces during the period October 14–November 5 (i.e., before the tier system);
- $a_r$ and $b_{r,p}$ are random effects, assumed to be normally distributed. $a_r$ allows random deviations from the mean among regions; $b_{r,p}$ allows random deviations from the mean regional mobility among provinces within a region;
- $\varepsilon_{p,T}$ is random noise assumed to be normally distributed.

**Changes in transmissibility**. To investigate the change in transmissibility observed after the introduction of tiers, we considered a pre/post-intervention design. We analyzed the mean SARS-CoV-2 transmissibility $R(t)$ over two time periods: October 30 to November 5 (i.e., before the regional tier system, when nationwide interventions were still in place) and November 19–25 (i.e., 2–3 weeks after the introduction of the tier system). We used LMM where regions and provinces were considered as nested random effects, to incorporate heterogeneous initial transmission levels across different geographical areas. All models included two independent variables (IV) and their interaction. The IVs were the time period (pre- or post-tiers) and a classification of the province/region based on the assigned tier. We considered five different models (Table 3): (A) the dependent variable (DV) was the mean value of $R(t)$ at the provincial level and the classification of regions was based on the maximum assigned tier (yellow, orange, or red) over the entire study period; (B) as (A), but considering as DV the log-transformed mean value of $R(t)$, so that the model estimates relative rather than absolute changes in $R(t)$ across periods; (C) as (A), but excluding provinces that changed tier assignment

during the study period, so that the grouping was based on the assigned tier only; (D) as (A), but the DVs and IVs are considered at the regional level; (E) as (A), but $R(t)$ values were estimated using a reduced serial interval with shape 1.14, rate 0.23 and mean 5 days, to assess the potential effect of active surveillance and contact tracing on the temporal dynamics of infection[24].

In addition to the five LMMs, we adopted an alternative modeling approach that did not require a grouping of regions in the category, to bypass potential biases associated with this operation. We considered a generative model based on the renewal equation[25] where we assumed that each tier would reduce $R(t)$ to a given value by means of a linear reduction over 14 days. The final value of R(t) for each tier was a free parameter and was calibrated by applying an MCMC procedure on the Poisson likelihood of observing the reported cumulative number of hospital admissions in each region. Full details for all models are reported in Supplementary Note 2.

**Changes in hospital admissions**. To evaluate the effect of the introduction of tiers on hospital burden, we projected for each region the curve of daily hospitalizations using the renewal equation[25] under the assumption that $R(t)$ would remain constant throughout the projection period and equal to the pre-tiers value. The estimated number of new hospital admission $H_i(t)$ in a region $i$ was thus given by

$$H_i(t) = \text{Pois}\left(R_i(t) \sum_{s=1}^{t} \varphi(s) h_i(t-s)\right) \qquad (3)$$

where

- $\text{Pois}(\lambda)$ is a Poisson sample with rate $\lambda$;
- $h_i(t)$ is the daily number of new hospital admissions in region $i$;
- $\varphi(s)$ is the distribution of the generation time discretized by day, as above;
- $R_i(t)$ is set for all $t$ to the mean value of $R(t)$ estimated in region $i$ in the period October 30–November 5, 2020.

We used as input for $h_i(t)$ the curve of daily hospital admissions until November 5, 2020 (i.e., the day before the enactment of the tiers) and projected hospital admissions from November 6 to 25. We simulated 1000 runs to take into account the stochastic variability of projections. All analyses were carried out using the statistical software R[26].

**Data ethics**. Google's Mobility data collected, aggregated, anonymized and shared by Google originated from users who have chosen to turn on the location history setting. Data used in the manuscript will remain available in a public repository (see "Data availability" section).

**Reporting summary**. Further information on research design is available in the Nature Research Reporting Summary linked to this article.

## Data availability
Mobility and epidemiological data have been deposited in figshare and are publicly available (https://doi.org/10.6084/m9.figshare.14153351.v1).

## Code availability
Code has been deposited in figshare and is publicly available (https://doi.org/10.6084/m9.figshare.14153351.v1).

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

## Acknowledgements
M.M., G.G., P.Po., V.M., F.T., P.Pe, F.R. and S.M. acknowledge funding from EU grant 874850 MOOD (cataloged as MOOD 021). The contents of this publication are the sole responsibility of the authors and do not necessarily reflect the views of the funders.

## Author contributions
G.G., S.B., P.Pezzotti, S.I., M.A., and S.M. designed research; M.M., V.M., G.G., A.V., F.R., A.M.U., P.Poletti, M.F., M.F.V., and X.A. performed research; M.M, V.M., G.G., A.V., M. d.M., A.B., P. Poletti, M.S., D.P, and F.T. analyzed data; and M.M, G.G., F.R., P. Poletti, F. T., P. Pezzotti, M.A., and S.M. wrote the paper. All authors contributed to data interpretation, critical revision of the manuscript and approved the final version of the manuscript.

## Competing interests
M.A. has received research funding from Seqirus. The funding is not related to COVID-19. All other authors declare no other competing interests.
