## [Peer Review File · Nature Communications]

REVIEWER COMMENTS

Reviewer #1 (Remarks to the Author):

This manuscript tries to estimate the impact of tiered restrictions on human activities and the epidemiology of the second wave of COVID-19 in Italy, using the linear mixed model with information of hospitalisation, epidemic and mobility data. This is definitely an important work and can help governments to assess their NPI's effectiveness. But I have several major issues:

(1) Parameter value issues: L205 and Line 258, the authors used the distribution of the serial interval, estimated from the analysis of contact tracing data in Lombard, with a gamma function with shape 1.87 and rate 0.28, having a mean of 6.6 days.

However, it has the high dynamics of the serial interval [Ali]. Do you think it is also true in Italy? And if yes, what is the impact of this dynamic on your results?

[Ali]: Serial interval of SARS-CoV-2 was shortened over time by nonpharmaceutical interventions, ST Ali, L Wang, EHY Lau, XK Xu, Z Du, Y Wu, GM Leung, BJ Cowling, Science 369 (6507), 1106-1109

(2) R_t estimation issue: L205, Infections in different states (symptomatic, pre-symptomatic and asymptomatic) may have different transmission rates. Do you think it is suitable to consider R_t specific for symptomatic individuals? And how do you think the impact of unreported symptomatic infections on your likelihood function in L205, given your data of hospital admissions.

(3) Table issues: Would you give R^2 to all your tables for results of linear mixed models? It is hard to know how much the data can be explained. And Can the stepwise regression be used in your linear model to exclude those unnecessary with larger p-value?

Reviewer #2 (Remarks to the Author):

This is a descriptive work of the course of the COVID-19 pandemic in Italy during the period where a three tiered approach to mitigation was adopted.

As measures were adopted at different times and different places, the authors try to estimate the impact of different approaches.

While I commend the authors for trying this analysis, there are major limitations in the methods and interpretation that are currently overlooked.

The largest problem in what is proposed is that the change in tiers in the regions was related, in part, to changes in what is modelled.

For example, change from yellow to orange was based on a reassessment of the epidemiological situation, which includes as stated change in cases or hospitalisation.

Regions that remained in yellow are those in which the intervention was not considered insufficient, while others went on to stricter measures.

There are therefore selection biases in the analysis: the parameter estimating change in the yellow

tier is conditional on the yellow intervention being successful/sufficient. It does not account for the regions in which the yellow tier was insufficient. As such, it is likely that it is biased towards being more efficient than it really was. This selection bias also affects the two other tiers.

It could explain why the "yellow tier" is just below 1 for R_t , as this is what would be expected for the region not to be reallocated to stricter measures.

An estimate of the "true" effect of yellow tier could be above 1 if one accounts for the regions where it was not successful and reinforced.

This limitation would in turn impact the estimation of avoided hospitalization, as it is based on the R_t estimated before.

There are also more traditional potential problems due to the "incomplete design" for the observation, as not all provinces were observed under all tiers. It is difficult to anticipate what could be the consequences of this, as the effects are estimated from different sets of provinces but essentially only from those with observation under different conditions.

As a whole, I believe that this seriously undermines the analyses presented here. Either it is possible to come up with a better analysis that will allow causal estimates as presented here, or the analysis should be presented as only descriptive. The interpretation in terms of avoided hospitalisations would have to be dropped.

Concerning the model, I wondered why $R(t)$ is estimated daily and later averaged. Wouldn't the "natural" model for the effect of changes under this kind of intervention be a step function for $R(t)$?

A parametrization for $R(t)$ as $R(t) = R(0) + \Delta R(t, \text{tier})$ could be more relevant and spare the averaging step. This could be in a sensitivity analysis.

The model for $R(t)$ also do not account for initial value and assume the same absolute reduction in all places.

I wondered whether the reduction in $R(t)$ was best modelled in absolute terms or should be relative. The current parametrization suggests that the decrease in R would be unconditional on the number of previous contacts, well it could be possible that a reduction by a given factor is better. This could be a sensitivity analysis.

There should be a random factor to account that some provinces are measured more than once in distinct tier levels, although this does not solve the issue. (so a c_{rpl} random parameter)

L47 : January *2021*

L75: "decreased progressively increasing with the tier level" could be "decreased progressively with increasing tier level"

In Table 2/ Fig 1; the caption should explain more directly the National / Tiers system, for example with (National - up to 11/5) Tiers (after 11/5)

Figure 3 : I did not see what the model fit demonstrates here. The model used amounts to computing the mean of the boxplots presented, with some subtleties in the weights.

It comes as no surprise that the red dots align with the data, it can't be otherwise and offers little validation.

Reviewer #3 (Remarks to the Author):

Review of "Impact of tiered restrictions on human activities and the epidemiology of the second wave of COVID-19 in Italy" by Manica et al.

In this paper, the authors propose to measure the effect of regionally-differentiated tiered restrictions during the second COVID-19 wave in Italy. This is a nice, short and clear paper on an important topic, and in particular the figures are really nice.

My major concern with this paper is that it's not clear to me that the measured effects on R_t are solely due to the interventions being assessed. From looking at Figure 2, it's not obvious to me that the impact of tiers per se can be clearly assessed from these data in the way that the authors are doing it, because decreases in R_t in most regions look like they are simply continuing a (mostly downward) trend established before November 6th.

R_t over the study period could be decreasing partly because of a build-up in herd immunity due to ongoing incidence. As the incidence was presumably higher in provinces that went into the red tier of restrictions, this could also partly explain why R_t decreased more in those provinces than in others.

In our paper in the Lancet Infectious Diseases on a similar topic ([https://www.thelancet.com/journals/laninf/article/PIIS1473-3099\(20\)30984-1/fulltext](https://www.thelancet.com/journals/laninf/article/PIIS1473-3099(20)30984-1/fulltext)), we tried to get around this issue in two ways.

First, we accounted for secular trends in mobility rates by subtracting a national trend in mobility from mobility indices before looking for changes due to tiered restrictions. This was much easier for us as it would be for the authors, as in England, our Tier 1 was essentially the same as what had come before, so it acted as a contemporaneous baseline for Tiers 2 and 3, which might not be as easy to apply to the situation in Italy. It may be the case for this study that only the comparative difference between yellow and orange and between yellow and red is measurable — after accounting for secular trends in mobility and R_t . In any event it would be good to see time series for mobility presented in the same way as the time series for R_t have been presented, so that if there is any apparent secular trend, this can be assessed by the reader.

Second, rather than measure R_t directly, we estimated changes in R by looking at the relationship between mobility rates and contact rates as measured in an ongoing contact survey. I know the authors may not have the data to follow this procedure exactly (and I am not requiring them to follow this procedure exactly), but without somehow accounting for secular trends in mobility and changes in R_t owing to the accumulation of herd immunity (unless the authors can argue that this was insignificant over the study period) then I think their estimates for the impact of tiers on R_t would be overestimates.

When looking at the estimates of the effects of tiers on mobility made by the authors (e.g. Figure 1), it seems clear that the impact of the tiers on mobility per se are much more modest than those estimated for R_t . I am not arguing that there is a linear relationship between mobility and R_t , but this to me supports the idea that the estimates of the impact of tiers on R_t , in particular, are overestimates.

Minor points

Results: We need a brief description in one or two sentences of how R_t was estimated and from what data in this section, as otherwise some later passages (e.g. on R_t estimated from hospital cases, and the potential impact of changes in testing rates) are difficult to understand.

“These levels of SARS-CoV-2 transmissibility accounted for the imposition of national restrictions since October 14” – this is a bit confusing to me. The R_t values are being measured from October 26, so how can they explain what was done on October 14?

Discussion: The introduction section provides a nice background to the epidemiology of the second wave in Italy. I would appreciate a brief discussion of what has happened since November 25 in Italy regarding the system of tiered restrictions (has it been kept in place or exchanged for something else)?

Methods:

Epidemiological data — Why is the upper limit on the summation over s the same as the upper limit on the product over t ? Also, instead of using $\phi(s)$, the probability density at time s , shouldn't the area under the probability density curve for the time period of interest, i.e. $\Phi(s) - \Phi(s - 1)$ where Φ is the cumulative distribution function, be used instead—or a discretised distribution as suggested by Cori et al (ref. 20)? Perhaps the authors could use the method of Cori et al. directly for estimating R_t ? I think readers would benefit from a much more detailed explanation of what the assumptions underlying this model are, including citations of previous work using similar methods (if available).

Impact of tiered restrictions on human activities — Since there are subscripts on M for the tier and province, it might be clearer to also include a subscript for the location category.

Impact on tiered restrictions on transmissibility — The authors use a linear mixed model for R_t estimations here, but it's not clear to me that this is the correct approach. Effects on R_t could be multiplicative, additive, or a mix of the two.

Reviewer #1 (Remarks to the Author):

This manuscript tries to estimate the impact of tiered restrictions on human activities and the epidemiology of the second wave of COVID-19 in Italy, using the linear mixed model with information of hospitalisation, epidemic and mobility data. This is definitely an important work and can help governments to assess their NPI's effectiveness. But I have several major issues:

We would like to thank the reviewer for taking time to evaluate our manuscript, for the positive evaluation of our work, and for providing constructive feedback. The manuscript has been extensively revised, and the main changes can be summarized as follows:

1. To increase robustness to possible different biases, we now report in the main text results from six different models. In particular, we included additional models where the relationship between $R(t)$ and tiers is not explored on the basis of a fixed grouping but rather taking into account tier-induced daily changes in $R(t)$, a different tier grouping is used, the effect of tiers on $R(t)$ is assumed to be multiplicative, and where we considered a shortened serial interval due to contact tracing and active surveillance.
2. To tone down claims of causality, we reduced the scope of our analysis on the incidence of hospitalizations, focusing only on comparing the observed hospital admissions against projections in which the reproduction number would remain equal to the one observed prior to the introduction of tiers.
3. We expanded our discussion to further comment on potential selection biases introduced by the study design (i.e., an ecological study), and we revised the wording used in both Results and Discussion.

Below we report a point-to-point response to all comments.

(1) Parameter value issues: L205 and Line 258, the authors used the distribution of the serial interval, estimated from the analysis of contact tracing data in Lombard, with a gamma function with shape 1.87 and rate 0.28, having a mean of 6.6 days.

However, it has the high dynamics of the serial interval [Ali]. Do you think it is also true in Italy? And if yes, what is the impact of this dynamic on your results?

[Ali]: Serial interval of SARS-CoV-2 was shortened over time by nonpharmaceutical interventions, ST Ali, L Wang, EHY Lau, XK Xu, Z Du, Y Wu, GM Leung, BJ Cowling, Science 369 (6507), 1106-1109

We thank the reviewer for raising this issue. Following the reviewer's comment, we performed an additional analysis to explore the effect of shortening the serial interval when estimating the impact of tiered restrictions on SARS-CoV-2 transmissibility. Using a previously published method [Marziano et al., PNAS 2021], we estimated that about 63% of infections between October 1, 2020, and January 15, 2021, remained undetected in Italy [Marziano et al., medRxiv 2021]. Therefore, we assumed that the serial interval could have been reduced to a maximum of two thirds (as in Ali et al) for at most 37% of the overall number of infections. The resulting distribution of the serial interval would arise from a mixture of two gamma distributions, the one used in the main analysis (shape = 1.87, rate = 0.28) with weight 0.63, and the one representing the shortened serial interval (shape = 1.08, rate = 0.48, mean = 2.2 days) with weight 0.37. This mixture distribution was well approximated by a gamma with shape = 1.14 and rate = 0.23, therefore resulting in a mean serial interval of 5 days, i.e., about 25% shorter than the baseline (see Appendix, figure reported below). This model (Model E) tended to estimate lower reductions in $R(t)$ associated to tiers, but generally in the same range as estimates from other models where a mean serial

interval of 6.68 days was used. These results are now reported in the main text and in Figure 3, and the full methodology is detailed in the Appendix.

(2) Rt estimation issue: L205, Infections in different states (symptomatic, pre-symptomatic and asymptomatic) may have different transmission rates. Do you think it is suitable to consider Rt specific for symptomatic individuals? And how do you think the impact of unreported symptomatic infections on your likelihood function in L205, given your data of hospital admissions.

We thank the reviewer for this comment that allowed us to clarify how estimates of $R(t)$ relate to different modeling assumptions. Although different clinical conditions may affect the individual transmission rate, the proportion of infections that are symptomatic can be assumed to remain approximately constant over time. Thus, the curve over time of symptomatic cases by date of symptom onset is a rescaled approximation of the curve of all infections, provided that the notification rate of symptomatic cases is approximately constant too. In particular, it has been shown that $R(t)$ estimates remain robust even when the reporting rate is very low or slightly fluctuating [Zhang et al, Lancet Inf Dis 2020; Liu et al, PLOS Comput Biol 2020]. Because temporal changes in the reporting of symptomatic cases cannot be completely excluded, we performed a sensitivity analysis where the net reproduction numbers were computed from the time series of hospital admitted cases (see Appendix), as we expect negligible changes in the hospitalization criteria during the observation period, and therefore in the reporting rate of cases requiring hospitalization. Results from this sensitivity analysis were consistent with the range of estimates provided in the main analysis. This point is discussed as follows:

“It is worth remarking that during periods of high infection incidence, the notification rate may decrease due to the saturation of tracing and testing capabilities. Although the method we adopted lead to estimates of $R(t)$ that are not affected by a constant underreporting rate⁹, we acknowledge that fluctuations in the reporting rate may alter $R(t)$ estimates¹⁰⁻¹². However, during the second wave of COVID-19 in Italy, the largest increase in the number of cases occurred in October; therefore, we expect the notification rate to have stabilized before the period considered for $R(t)$ (October 30 – November 25). In addition, hospital admission rates are less subject to changes compared with notification rates of symptomatic cases, and we found similar results when using transmissibility estimates computed from the curve of hospital admissions (Appendix).”

(3) Table issues: Would you give R2 to all your tables for results of linear mixed models? It is hard to know how much the data can be explained. And Can the stepwise regression be used in your linear model to exclude those unnecessary with larger p-value?

We now added the R2 for linear mixed models of the main analysis in Table 3 and report its value for additional sensitivity analyses in the Appendix. The coefficient of determination R2 for mixed models was computed following the approach described in Nakagawa et al., (J Royal Soc Interf, 2017) and the resulting values ranged between 0.57 and 0.70 for models in the main analysis.

Following the reviewer's suggestion, we also adopted a stepwise model selection procedure based on the likelihood ratio tests to define the optimal set of covariates for our main analysis. The performed analysis confirmed the appropriateness of our original formulation, showing that the model $Y_{p,T} = \beta_0 + \beta_1 X_p^{orange} + \beta_2 X_p^{red} + \beta_3 Z_T + \beta_4 X_p^{orange} Z_T + \beta_5 X_p^{red} Z_T$ should be preferred to all models nested in this formulation. Likelihood ratio tests on the comparison between the full model and nested models always resulted in p-values < 0.0001 (see Table S13, which is reported below for reviewer convenience). The adopted selection procedure is now described in the Appendix.

Model	Chisq	DF	pvalues
$Y_{p,T} = \beta_0 + \beta_1 X_p^{orange} + \beta_2 X_p^{red} + \beta_3 Z_T$ (tier+period)	27.656	2	<0.0001
$Y_{p,T} = \beta_0 + \beta_1 X_p^{orange} + \beta_2 X_p^{red}$ (tier)	203.91	3	<0.0001
$Y_{p,T} = \beta_0 + \beta_3 Z_T$ (period)	27.904	4	<0.0001

Reviewer #2 (Remarks to the Author):

This is a descriptive work of the course of the COVID-19 pandemic in Italy during the period where a three tiered approach to mitigation was adopted. As measures were adopted at different times and different places, the authors try to estimate the impact of different approaches. While I commend the authors for trying this analysis, there are major limitations in the methods and interpretation that are currently overlooked.

We would like to thank the reviewer for taking time to evaluate our manuscript and for providing constructive feedback. The manuscript has been extensively revised, and the main changes can be summarized as follows:

1. To increase robustness to possible different biases, we now report in the main text results from six different models. In particular, we included additional models where the relationship between $R(t)$ and tiers is not explored on the basis of a fixed grouping but rather taking into account tier-induced daily changes in $R(t)$, a different tier grouping is used, the effect of tiers on $R(t)$ is assumed to be multiplicative, and where we considered a shortened serial interval due to contact tracing and active surveillance.
2. To tone down claims of causality, we reduced the scope of our analysis on the incidence of hospitalizations, focusing only on comparing the observed hospital admissions against projections in which the reproduction number would remain equal to the one observed prior to the introduction of tiers.
3. We expanded our discussion to further comment on potential selection biases introduced by the study design (i.e., an ecological study), and we revised the wording used in both Results and Discussion.

Below we report a point-to-point response to all comments.

The largest problem in what is proposed is that the change in tiers in the regions was related, in part, to changes in what is modelled. For example, change from yellow to orange was based on a reassessment of the epidemiological situation, which includes as stated change in cases or hospitalisation. Regions that remained in yellow are those in which the intervention was not considered insufficient, while others went on to stricter measures. There are therefore selection biases in the analysis: the parameter estimating change in the yellow tier is conditional on the yellow intervention being successful/sufficient. It does not account for the regions in which the yellow tier was insufficient. As such, it is likely that it is biased towards being more efficient than it really was. This selection bias also affects the two other tiers. It could explain why the "yellow tier" is just below 1 for R_t , as this is what would be expected for the region not to be reallocated to stricter measures. An estimate of the "true" effect of yellow tier could be above 1 if one accounts for the regions where it was not successful and reinforced.

As correctly pointed out by the reviewer, tiers were (and are) assigned to Italian regions on the basis of a wide spectrum of epidemiological indicators, including but not limited to the reproduction number. It is worth mentioning that surveillance data for the estimates of $R(t)$ suffer from intrinsic diagnostic and reporting delays. As a consequence, the estimates of the reproduction number available to decision-makers at each assessment refer to the epidemiological situation of 14 days before. Moreover, of 12 tier changes occurring in the study period, 7 occurred within 4 days from the previous assignment, and 11 within 14 days from the introduction of tiers. Thus, we argue that most of these changes were not decided on the basis of an evaluation of insufficient effectiveness, for the evaluation of which in terms of $R(t)$ there was insufficient time, but because of incoming updates on the value of $R(t)$ from 14 days before. Furthermore, the tier assignments (and their updates) were not decided solely based on the value of $R(t)$. Some regions were assigned to the most restrictive tier (red) as a

consequence of incompleteness in reported data, which was interpreted by governmental officials as a lack of resilience by the regional health system in monitoring the epidemiological situation. In other cases, even when incidence levels had stabilized (e.g., $R(t)$ close to 1), more restrictive measures may have been imposed to decrease the incidence of infection and reduce the pressure on local health care systems. We apologize for the lack of information on these important points that may have caused confusion, and we are grateful for this comment that allowed us to amend the manuscript by adding these key pieces of information.

“Tiers were assigned to each of the 21 regions and autonomous provinces (AP) after an epidemiological risk assessment by the Ministry of Health, based on the combination of a twenty-one quantitative indicators on: i) the level of transmission, ii) the burden on older age groups and healthcare, and iii) resilience of monitoring systems. [...] Due to the intrinsic delays in the dynamics of infection, healthcare seeking, and data collection, estimates of the reproduction number (measuring the level of transmission) available at each epidemiological assessment were relative to 14 days before the date of assessment. To guarantee a prompt reaction to epidemiological updates, the assignment of tiers was therefore re-evaluated according to frequent reassessments of incoming data from regions.”

To address the reviewer’s concerns, we now include results from an additional model based on the renewal equation (reported as Model F in the main text) where the potential impact of tiers is estimated by explicitly simulating the daily reduction of $R(t)$ associated to each specific day spent in a given tier. This allows us to avoid the assignment of regions to a specific grouping, and to factor in the cumulative effect of changes in tier assignment over time. Estimates on the final $R(t)$ expected after two weeks in a given tier were consistent with equivalent results from five alternative statistical models. We acknowledge that any possible effort to estimate the effect of tiered restrictions in Italy is affected by the lack of a case-control study design. We think is an issue shared by all analyses focusing on intervention measures, since measures are necessarily adapted to time-varying epidemiological conditions to serve public health needs. Despite the inevitable limitations and potential biases characterizing our study design, we believe that the quantification of the association between tiers and transmissibility represents an important piece of information to monitoring and response activities.

To recognize the limitation properly pointed out by the reviewer, we have included the following paragraph in the discussion:

“Finally, our estimates are subject to further potential biases related to the ecological study design; therefore, figures associated to each tier should be taken with caution, despite their robustness across a number of modeling assumptions. For example, regions that experienced an escalation in their assigned tier over the study period might have altered our estimates for the effect of lower tier(s) if they were not reassigned, thus potentially biasing the estimate upward.”

This limitation would in turn impact the estimation of avoided hospitalization, as it is based on the R_t estimated before. There are also more traditional potential problems due to the "incomplete design" for the observation, as not all provinces were observed under all tiers. It is difficult to anticipate what could be the consequences of this, as the effects are estimated from different sets of provinces but essentially only from those with observation under different conditions. As a whole, I believe that this seriously undermines the analyses presented here. Either it is possible to come up with a better analysis that will allow causal estimates as presented here, or the analysis should be presented as only descriptive. The interpretation in terms of avoided hospitalisations would have to be dropped.

In light of the potential biases, we toned down all claims of causality for the effect of tiers throughout the manuscript. As suggested, we dropped this analysis. Instead, we show only the

simulation results on averted hospitalizations against a counterfactual scenario where daily hospital admissions are projected under the assumption that the existing average $R(t)$ in the week October 30-November 5 would be maintained for the whole study period (Figure 4). We believe this result does not involve any assumptions on the effectiveness of interventions but only compares the observed data to those expected in a status-quo scenario.

Concerning the model, I wondered why $R(t)$ is estimated daily and later averaged. Wouldn't the "natural" model for the effect of changes under this kind of intervention be a step function for $R(t)$? A parametrization for $R(t)$ as $R(t) = R(0) + \Delta R(t, \text{tier})$ could be more relevant and spare the averaging step. This could be in a sensitivity analysis.

Estimates of the reproduction numbers are usually computed as a rolling average of daily estimates [Cori et al, AJE 2013] because the latter are severely affected by the stochastic noise in the data. Throughout the paper, we refer to estimates computed as a weekly rolling average, in accordance with official values that are also used for public health decisions in Italy, including the assignment of tiers. Because of this, estimates of $R(t)$ are generally not characterized by sudden drops, but by smoother variations, even when the underlying transmissibility changes instantaneously. To clarify this point, we provide an illustrative example. We simulate a synthetic time series of cases (epidemic curve) using the renewal equation, assuming $R(t)=2$ for the first 35 days and then $R(t)=0.8$ in the following 35 days; we then estimate $R(t)$ from the synthetic epidemic curve. In the figure below, the upper panel shows the synthetic epidemic curve; the lower panel shows the theoretical $R(t)$ (black dashed line), the raw daily estimate (orange line: mean; shaded area: 95%CI), and its weekly rolling average (green line and shaded areas), computed using the methodology originally developed by Cori et al. [AJE, 2013].

In practice, previous studies [Guzzetta et al., EID 2021] have demonstrated that at least two weeks were required to fully appreciate the reduction of $R(t)$ in Italian regions after the hard lockdown following the first COVID-19 wave in spring 2020, likely due to other kinds of inertia in the epidemic dynamics. For this reason, we considered the estimates of $R(t)$ one week before the introduction of tiers (Oct 30-Nov 5, 2020) and in the week starting 14 days after (Nov 19-25, 2020). This rationale is also supporting our formulation of gradual, rather than sudden, changes in $R(t)$ within Model F.

The model for $R(t)$ also do not account for initial value and assume the same absolute reduction in all places. I wondered whether the reduction in $R(t)$ was best modelled in absolute terms or should be relative. The current parametrization suggests that the decrease in R would be unconditional on the number of previous contacts, well it could be possible that a reduction by a given factor is better. This could be a sensitivity analysis. There should be a random factor to account that some provinces are

measured more than once in distinct tier levels, although this does not solve the issue. (so a `c_rpl` random parameter).

We apologize for the lack of clarity on this point. In the five considered LMMs (models A-E in Table 3 and Figure 3), each province/region is always measured only once in a given tier grouping. For example, provinces of Tuscany, whose tier escalated from yellow to orange to red, were always considered in the red tier for models where regions are grouped by maximum tier (A, B, D, E) and were excluded from the analysis in model C, where we only considered regions that maintained the same tier over the study period. In order to account for heterogeneous transmission levels across different geographical areas, a random effect term for the region and a nested random effect for different provinces within the same region have been considered in all LMMs. Following the reviewer's suggestion, for which we are grateful, one of the models included in the main analysis (Model B) now explores relative rather than net changes in $R(t)$. We have deeply revised the Methods to provide an overview of all adopted models and report full methodological details for all models in the Appendix. We added a more explicit note on provincial and regional random effects in the text as follows:

"We used linear mixed models (LMM) where regions and provinces were considered as nested random effects, to incorporate heterogeneous initial transmission levels across different geographical areas."

L47 : January *2021*

Addressed

L75: "decreased progressively increasing with the tier level" could be "decreased progressively with increasing tier level"

Addressed

In Table2/Fig 1; the caption should explain more directly the National / Tiers system, for example with (National - up to 11/5) Tiers (after 11/5).

Following the reviewer's suggestion, we included in the captions of Table 2 and of Figure 1 the following sentence:

"National interventions were in place from October 14 to November 5, 2020, while values for different tiers refer to the period November 6-November 25."

Figure 3 : I did not see what the model fit demonstrates here. The model used amounts to computing the mean of the boxplots presented, with some subtleties in the weights. It comes as no surprise that the red dots align with the data, it can't be otherwise and offers little validation.

Figure 3 has now been replaced by a comparison of estimates obtained by LMMs.

Reviewer #3 (Remarks to the Author):

Review of “Impact of tiered restrictions on human activities and the epidemiology of the second wave of COVID-19 in Italy” by Manica et al.

In this paper, the authors propose to measure the effect of regionally-differentiated tiered restrictions during the second COVID-19 wave in Italy. This is a nice, short and clear paper on an important topic, and in particular the figures are really nice.

We would like to thank the reviewer for taking time to evaluate our manuscript, for the positive evaluation of our work, and for providing constructive feedback. The manuscript has been extensively revised, and the main changes can be summarized as follows:

1. To increase robustness to possible different biases, we now report in the main text results from six different models. In particular, we included additional models where the relationship between $R(t)$ and tiers is not explored on the basis of a fixed grouping but rather taking into account tier-induced daily changes in $R(t)$, a different tier grouping is used, the effect of tiers on $R(t)$ is assumed to be multiplicative, and where we considered a shortened serial interval due to contact tracing and active surveillance.
2. To tone down claims of causality, we reduced the scope of our analysis on the incidence of hospitalizations, focusing only on comparing the observed hospital admissions against projections in which the reproduction number would remain equal to the one observed prior to the introduction of tiers.
3. We expanded our discussion to further comment on potential selection biases introduced by the study design (i.e., an ecological study), and we revised the wording used in both Results and Discussion.

Below we report a point-to-point response to all comments.

My major concern with this paper is that it's not clear to me that the measured effects on R_t are solely due to the interventions being assessed. From looking at Figure 2, it's not obvious to me that the impact of tiers per se can be clearly assessed from these data in the way that the authors are doing it, because decreases in R_t in most regions look like they are simply continuing a (mostly downward) trend established before November 6th. R_t over the study period could be decreasing partly because of a build-up in herd immunity due to ongoing incidence. As the incidence was presumably higher in provinces that went into the red tier of restrictions, this could also partly explain why R_t decreased more in those provinces than in others.

Between November 6 and November 25, 2020, 656,044 infections were notified in Italy, representing approximately 1.1% of the total population. Even in more affected regions, like Lombardy or Piedmont, this percentage did not exceed 1.6%. Using a previously published method [Marziano et al. PNAS 2021], we estimated that about 37% of all infections were notified in Italy between October 1, 2020 and January 15, 2021 [Marziano21, medRxiv]. Thus, we expect that natural immunity acquired in the considered period would represent at most 3% of the Italian population (and 5% of that of Lombardy and Piedmont). Therefore, the reviewer is correct in suggesting that the accumulated immunity may have had an impact on reducing SARS-CoV-2 transmissibility over the study period, although with a minor contribution. We thank the reviewer for this useful suggestion; we have added the following sentence to the discussion to acknowledge the effect of the accumulation of immunity in the population:

“It is important to consider that a part of the estimated reduction in reproduction numbers have resulted from the accumulation of natural immunity. Between November 6 and November 25, 2020, the cumulative notified cases represented 1.1% of the total Italian

population. Considering a case notification ratio of 37%¹³, we estimate a 3% depletion of the susceptible reservoir in Italy (with range 1-5% across Italian regions), translating into similar reductions of $R(t)$ due to the accumulated immunity.”

In our paper in the Lancet Infectious Diseases on a similar topic ([https://www.thelancet.com/journals/laninf/article/PIIS1473-3099\(20\)30984-1/fulltext](https://www.thelancet.com/journals/laninf/article/PIIS1473-3099(20)30984-1/fulltext)), we tried to get around this issue in two ways. First, we accounted for secular trends in mobility rates by subtracting a national trend in mobility from mobility indices before looking for changes due to tiered restrictions. This was much easier for us as it would be for the authors, as in England, our Tier 1 was essentially the same as what had come before, so it acted as a contemporaneous baseline for Tiers 2 and 3, which might not be as easy to apply to the situation in Italy. It may be the case for this study that only the comparative difference between yellow and orange and between yellow and red is measurable — after accounting for secular trends in mobility and R_t . In any event it would be good to see time series for mobility presented in the same way as the time series for R_t have been presented, so that if there is any apparent secular trend, this can be assessed by the reader. When looking at the estimates of the effects of tiers on mobility made by the authors (e.g. Figure 1), it seems clear that the impact of the tiers on mobility per se are much more modest than those estimated for R_t . I am not arguing that there is a linear relationship between mobility and R_t , but this to me supports the idea that the estimates of the impact of tiers on R_t , in particular, are overestimates.

We thank the reviewer for this suggestion. To explore the potential role played by secular trends in determining the observed changes in transmissibility, we have updated Figure 2 (reported below for the reviewer’s convenience). Thin lines represent raw data for each province, thick straight lines are regression lines superimposed to raw data to help the visual identification of trends.

The figure highlights three points:

- 1) when restrictions were constant over time (i.e., before October 14 and after November 6), mobility trends were approximately flat in all categories, except for some waning over time of mobility reductions in the red tier (likely due to a reduced compliance with stay-

home orders). Between October 14 and November 6, a marked linear trend in mobility indicators may be attributed to the progressive increase in restrictions with spatially and temporally heterogeneous application (see Table 1, column National restrictions). These national restrictions may explain the declining trends in $R(t)$ before the introduction of tiers, shown in Figure 2 for the period October 30-November 5;

- 2) the impact of tiers on human mobility was instantaneous and quite important; the significant changes in mobility introduced by tier are shown in Table 2 and are of similar magnitude as the estimates for the reproduction numbers, although a comparison is difficult since Google mobility indicators are provided as relative values;
- 3) a significantly higher decrease in human activity levels was observed in provinces where more restrictive tiers were adopted.

This observation supports our assumption that secular trends were not very relevant for the case of Italy and that the largest part of the variation in human activities was due to changing restrictions. We have now rewritten most of the section “Changes in human activities after the introduction of tiers” to provide these observations and better support the absence of secular trends in human activities before interventions.

Second, rather than measure R_t directly, we estimated changes in R by looking at the relationship between mobility rates and contact rates as measured in an ongoing contact survey. I know the authors may not have the data to follow this procedure exactly (and I am not requiring them to follow this procedure exactly), but without somehow accounting for secular trends in mobility and changes in R_t owing to the accumulation of herd immunity (unless the authors can argue that this was insignificant over the study period) then I think their estimates for the impact of tiers on R_t would be overestimates.

We believe that the two points above about the accumulation of herd immunity and secular trends in mobility support the robustness of our study. Overall, we toned down all claims of causality related to our estimates and we added the following comment in the discussion to highlight the important limitation related to the study design.

“Our estimates are subject to potential biases related to the ecological study design; therefore, figures associated to each tier should be taken with caution, despite their robustness across a number of modeling assumptions.”

Minor points

Results: We need a brief description in one or two sentences of how R_t was estimated and from what data in this section, as otherwise some later passages (e.g. on R_t estimated from hospital cases, and the potential impact of changes in testing rates) are difficult to understand.

We thank the reviewer for this suggestion. The following remark has been added at the beginning of the subsection entitled “Changes in transmissibility”:

“As a measure of transmissibility, we considered the net reproduction number $R(t)$, estimated from the epidemic curve of symptomatic cases by date of symptom onset, given a distribution of the serial interval.”

“These levels of SARS-CoV-2 transmissibility accounted for the imposition of national restrictions since October 14” – this is a bit confusing to me. The R_t values are being measured from October 26, so how can they explain what was done on October 14?

We apologize for the incorrect wording. We meant that those values were incorporating the effects of national restrictions imposed since October 14. However, the sentence has now been removed from the text.

Discussion: The introduction section provides a nice background to the epidemiology of the second wave in Italy. I would appreciate a brief discussion of what has happened since November 25 in Italy regarding the system of tiered restrictions (has it been kept in place or exchanged for something else)?

To address the reviewer's comment the following sentence was added in the Discussion:

“As of May 2021, the tier system is still in place, with limited changes to the enacted restrictions, and represents the key strategy to control and mitigate transmission in Italy as vaccination rolls out.”

Methods:

Epidemiological data — Why is the upper limit on the summation over s the same as the upper limit on the product over t ? Also, instead of using $\phi(s)$, the probability density at time s , shouldn't the area under the probability density curve for the time period of interest, i.e. $\Phi(s) - \Phi(s - 1)$ where Φ is the cumulative distribution function, be used instead—or a discretised distribution as suggested by Cori et al (ref. 20)? Perhaps the authors could use the method of Cori et al. directly for estimating R_t ? I think readers would benefit from a much more detailed explanation of what the assumptions underlying this model are, including citations of previous work using similar methods (if available).

We thank the reviewer for the careful reading of our manuscript. As correctly suggested, the upper limit on the product over s should be “ t ” (instead of “ T ” as previously written), and $\phi(s)$ represents the integral between time s and $s-1$ of the probability density describing the serial interval distribution. We revised the text accordingly.

Impact of tiered restrictions on human activities — Since there are subscripts on M for the tier and province, it might be clearer to also include a subscript for the location category.

We apologize for the lack of clarity. Separate regression models were applied to each location category. As such, we think that the inclusion of a subscript identifying the location may suggest the misleading idea that data from all locations were considered at once in a single model. To clarify the adopted approach, we change the text introducing the model equation as follows:

“To assess the impact of tiers on human activities, we used mobility data to calibrate a linear mixed model for each of the location categories in the Google reports. The linear mixed models are of the form: [...]”

Impact on tiered restrictions on transmissibility — The authors use a linear mixed model for R_t estimations here, but it's not clear to me that this is the correct approach. Effects on R_t could be multiplicative, additive, or a mix of the two.

We thank the reviewer for this useful suggestion. We agree with the reviewer that alternative effects of tiered restrictions on transmissibility should have been explored. Following her/his comment, we now report in the main text results of a new analysis where relative rather than net changes in $R(t)$ are explored (Model B in Table 3 and Figure 3), using an LMM on log-transformed values.

REVIEWERS' COMMENTS

Reviewer #1 (Remarks to the Author):

No more questions. Thanks for your detailed analysis for all of my comments.

Reviewer #2 (Remarks to the Author):

I think the authors did a good job of trying to discuss the issues raised.

Reviewer #3 (Remarks to the Author):

The authors have satisfactorily responded to my previous comments. This is a nice paper.

I just have one more very minor suggestion, which is that the methodology would be a bit clearer if the authors clarify that the 21 "regions and autonomous provinces" are what is referred to as the "regional level" in the paper, and that the provincial level is then a subdivision of this into 107 provinces. I think just a minor change in wording around line 95 (before Figure 2 is mentioned) would work, or around line 58. The potential confusion arises because line 95's reference to "by region and province" is easily confused with the earlier reference to "regions and autonomous provinces" in line 58, though line 58 is specifying that the Aosta Valley is both a region and a province, while line 95 is differentiating between the regional level (21 subdivisions) and the provincial level (107 subdivisions).

REVIEWERS' COMMENTS

Reviewer #1 (Remarks to the Author):

No more questions. Thanks for your detailed analysis for all of my comments.

Reviewer #2 (Remarks to the Author):

I think the authors did a good job of trying to discuss the issues raised.

Reviewer #3 (Remarks to the Author):

The authors have satisfactorily responded to my previous comments. This is a nice paper.

We would like to thank the reviewer for their suggestions and comments that greatly improved the quality of our manuscript.

I just have one more very minor suggestion, which is that the methodology would be a bit clearer if the authors clarify that the 21 "regions and autonomous provinces" are what is referred to as the "regional level" in the paper, and that the provincial level is then a subdivision of this into 107 provinces. I think just a minor change in wording around line 95 (before Figure 2 is mentioned) would work, or around line 58. The potential confusion arises because line 95's reference to "by region and province" is easily confused with the earlier reference to "regions and autonomous provinces" in line 58, though line 58 is specifying that the Aosta Valley is both a region and a province, while line 95 is differentiating between the regional level (21 subdivisions) and the provincial level (107 subdivisions).

We thank the reviewer for requesting this clarification that may indeed be confusing for a non-Italian reader. As correctly noticed by the reviewer the 21 regions and autonomous provinces are what is referred to as the "regional level" in the paper, and that the provincial level is then a subdivision of this into 107 provinces. To avoid confusion, included the following statement: "We considered changes in transmissibility at the regional and provincial level. The regional level comprises 19 regions plus the Autonomous Provinces (APs) of Trento and Bolzano, while the provincial level comprises 107 provinces (including the two APs)." and revised line 95 as follows: "were highly variable both at regional and provincial level" as in Figure 2 are included all 21 regions and autonomous provinces (regional level, one in each panel) as well as all the subdivision in 107 provinces (provincial level) distributed accordingly within each panel.